# Use of paclitaxel carried in solid lipid nanoparticles to prevent peritoneal fibrosis in rats

**Filipe M. O. Silva**[1,2]**, Priscila O. Carvalho**[3]**, Elerson C. Costalonga**[1]**, Rafael Pepineli**[1,2]**, Raul C. Maranhão**[3,4]*****, Irene L. Noronha**[1]

**1** Laboratorio de Nefrologia Celular, Genetica e Molecular, Divisao de Nefrologia, Faculdade de Medicina, Universidade de Sao Paulo, Sao Paulo, São Paulo, Brazil, **2** Departmento de Ciencias Medicas, Universidade Nove de Julho, Sao Paulo, São Paulo, Brazil, **3** Instituto do Coracao (InCor), Hospital das Clinicas HCFMUSP, Faculdade de Medicina, Universidade de Sao Paulo, Sao Paulo, São Paulo, Brazil, **4** Faculdade de Ciencias Farmaceuticas, Universidade de Sao Paulo, Sao Paulo, São Paulo, Brazil

* ramarans@usp.br

## Abstract

**Data Availability Statement:** All relevant data are within the manuscript and its Supporting information files.

### Background

Progressive fibrous thickening of peritoneal membrane (PM) is a major complication of long-term peritoneal dialysis. TGF-β/SMAD pathway activation, inflammation and neoangiogenesis have an important role in PM changes induced by peritoneal dialysis. Here, we investigated the effects of paclitaxel (PTX) carried in lipid core nanoparticles (LDE) on the development of peritoneal fibrosis (PF) in rats.

### Methods

To induce PF, 21 male Wistar rats (300-350g) were injected with chlorhexidine gluconate for 15 consecutive days and randomly assigned to three groups: 1)PF, n = 5: no treatment; 2) LDE, n = 8: treated with LDE only, 3/3 days during 15 days; 3)LDE-PTX, n = 8: treated with PTX (4mg/kg) associated with LDE, 3/3 days during 15 days. A Control group without PF induction (n = 5) was designed, received saline solution, 3/3 days. Peritoneum function tests were performed, and anterior abdominal wall samples of the PM were collected for analyses of peritoneal thickness, immunohistochemistry, and gene expression.

### Results

LDE-PTX treatment preserved the membrane function, maintaining the ultrafiltration rate and mass transfer of glucose at normal levels. LDE-PTX also prevented PM thickening induced by chlorhexidine gluconate injections. LDE-PTX treatment reduced the number of myofibroblasts infiltrating PM and inhibited the cell proliferation. Gene expression of fibronectin, FSP-1, VEGF, TGF-β, and SMAD3 were reduced by LDE-PTX.

**Funding:** This study was supported by the State of Sao Paulo Research Support Foundation (FAPESP, Sao Paulo, Brazil, Grant number 2014/03742-0) and by the National Institute of Science and Technology for Complex Fluids of the Ministry of Science, Technology and Innovation of Brazil.

**Competing interests:** The authors have declared that no competing interests exist.

## Conclusions

LDE-PTX was effective to prevent development of PF and preserve the PM filtration capacity in this rat model, with clear-cut actions on pro-fibrotic mechanisms. Thus, LDE-PTX can be candidate for future clinical trials as adjuvant to peritoneal dialysis to prevent PF development, since this preparation is devoid of toxicity as shown previously.

## Introduction

Despite the predominant use of hemodialysis as renal replacement therapy, which accounts for 90% of the procedures in the U.S.A. [1], Europe [2] and Brazil [3], the superiority of hemodialysis over peritoneal dialysis (PD) for patients with end-stage renal disease is debatable. Recent studies have shown that the relative mortality risk of patients undergoing PD is lower compared to patients on hemodialysis, especially in the first two years on renal replacement therapy [4]. Conversely, the less expensive, home-based PD procedure, which is more often used in lower- and middle-income countries, has important shortcomings. Long term exposure to PD fluids induce inflammation, peritonitis, neoangiogenesis and fibrosis of peritoneal membrane [5, 6] that may ultimately result in PD failure [6–8]. In addition, some patients develop encapsulating peritoneal sclerosis, a rare condition of excessive peritoneal fibrosis which is accompanied by high mortality rates [9, 10]. Currently, we have few options to treat this condition.

Over recent years, transforming growth factor-β (TGF-β) has been pointed out as a key fibrogenic factor involved in peritoneal fibrosis [11–13]. Activation of TGF-β by receptors on cell surface leads to downstream activation of SMAD proteins so that TGF-β/SMAD signalling has been recognized as an important pathway in the development of peritoneal fibrosis [14]. Blocking this pathway using anti-fibrotic and anti-proliferative molecules has been proposed as a therapeutic target.

Paclitaxel (PTX) is a potent anti-proliferative agent used in cancer treatment. PTX inhibits cell division by polymerization of tubulin, leading to the formation of dysfunctional microtubules [15]. The high toxicity of PTX is a major limiting factor as a long-term therapeutic option for eventual use to treat peritoneal fibrosis. However, when associated with solid lipid core nanoparticles (LDE), PTX toxicity is virtually abolished and thereby LDE-PTX may offer a new tool for the treatment of degenerative diseases that involve cell proliferation and chronic inflammation. LDE is taken up by low-density lipoprotein (LDL) receptors that are overexpressed in cells with increased mitosis rates. In those cells, increased uptake of cholesterol is required for cell membrane synthesis. Thus, LDL receptor upregulation is the mechanism for directing drugs to target sites by incorporating those drugs to LDE nanoparticles [16, 17].

The ability of LDE to drastically reduce the toxicity of PTX was shown not only in studies in experimental animals [16, 18, 19], but also in patients with advanced cancers [17, 20, 21] and patients with cardiovascular disease [22]. LDE-PTX association was shown stable in the bloodstream of animals and human patients and the pharmacological action is enhanced by this formulation compared with the commercial formulation of PTX in which this agent is carried with Cremophor EL [16].

Increased cellular proliferation, inflammation and fibrosis of the peritoneal membrane are complications of long-term peritoneal dialysis. The experimental model that most closely resembles this clinical situation is that of the animals chronically submitted to periodic infusions and effusions of dialysis solutions utilizing peritoneal catheters. However, catheter

malfunction and infection represent important bias for the interpretation of results and lead to great animal loss. Other models, such as mechanical injury by scraping the peritoneum and the use of 4.25% peritoneal dialysis solution require anesthesia and frequently do not produce the desired intensity of peritoneal fibrosis. In this setting, the induction of peritonitis by intraperitoneal chlorhexidine gluconate injections is a reliable and practical model that requires short periods for lesion development, with low animal mortality rates [23, 24].

Since lipid nanoparticles associated with PTX have anti-inflammatory and anti-fibrotic effects, this study aimed to investigate in rats whether LDE-PTX could protect against peritoneal fibrosis induced by intraperitoneal injections of chlorhexidine gluconate.

## Material and methods

### Animal model and experimental groups

Twenty-six adult male Wistar rats, weighing 300 to 350 g, obtained from an established colony at the University of Sao Paulo, Brazil, were used in this study. Rats were maintained in rodent cages in a 22°C room with a 12-h light-dark cycle with standard rat chow and water *ad libitum*. All animals were handled based on the guidelines for use, maintenance, and welfare of animals, established by the Brazilian federal law on the scientific use of animals (Law 1.1794/2008) and the Normative Resolution of the National Council for the Control of Animal Experimentation-CONCEA). All experimental procedures conducted in this study were previously approved by the Animal Use Ethics Committee of the Medical School Hospital of the University of the Sao Paulo (HCFMUSP), number 460/11.

Peritoneal fibrosis (PF) was induced by daily intraperitoneal (IP) injections of chlorhexidine gluconate (CG) at 0.1% in 15% ethanol dissolved in saline for 15 consecutive days.

Animals were randomly allocated to four groups:

1. PF (n = 5): rats were submitted to CG-induced PF.

2. LDE (n = 8): rats were submitted to CG-induced PF and treated with LDE only (IP), every 3 days from the first day of CG injection until day 15.

3. LDE-PTX (n = 8): rats were submitted to CG-induced PF and treated with PTX (4 mg/kg, IP) associated with LDE, every 3 days from the first day of CG injection until day 15.

4. Controls (n = 5): rats without PF induction and received saline solution every 3 days until day 15.

On day 15, peritoneum function tests were performed, and then, animals were euthanized with IP sodium pentobarbital (100 mg/Kg). Anterior abdominal wall samples of the peritoneal membrane were collected for posterior analysis.

### LDE preparation and association with Paclitaxel (LDE-PTX)

To increase the stability and yield of the LDE-PTX mixture, a PTX derivative, paclitaxel oleate compound, was synthesized. For derivatization, PTX (Pharmaceuticals, Shangai, China), was diluted in anhydrous dichloromethane and to this solution was added oleic acid, DCC and DMAP. The mixture was stirred under an inert atmosphere for 2h at room temperature [16]. The reaction was monitored by ultrahigh-performance liquid chromatography (UHPLC) (Nexera X2 Shimadzu, MO, USA) to analyze the conversion rate of the PTX.

LDE was prepared with lipid mixtures consisting of 55% phosphatidylcholine (Lipoid, Ludwigshafen, Germany), 25% esterified cholesterol (Aesar, MA, USA), 1% non-esterified cholesterol (Fabrichem, CT, USA), and 19% triglycerides (Migyol 812N, Sasol Germany GmbH,

Hamburg, Germany) [25]. To prepare LDE-PTX, PTX derivative was added to the mixture of lipids at a drug: lipid ratio 1:10. Emulsification of lipids was performed in a high-pressure homogenizer Emulsiflex C5 (Avestin, Ottawa, Canada). The particle size (50–70 nm) was measured by the dynamic light scattering method at a 90˚ angle using the ZetaSizer Nano ZS90 equipment (Malvern, Malvern, UK). The association rate of the drug to LDE was analyzed by UHPLC (Nexera X2) using the low-pressure gradient method on C18 silica column, mobile phase methanol 100%, and UV-visible detector, at 227 nm. The nanoparticles were sterilized by a polycarbonate membrane filter with 0.22 μm pore diameter (Merck Millipore, MA, USA) in laminar flow and stored at 4ºC in sterile glass bottles.

### Peritoneal function test

On day 15, before the euthanasia, 0.09 mL/g body weight of 4.25% peritoneal dialysis solution (Fresenius Medical Care, SP, Brazil) was administered IP and 2h later, the peritoneum was opened and the fluid was removed for ultrafiltration measurement. The ultrafiltration value was considered the volume of fluid removed after 2h minus the volume of fluid administered. After, peritoneal fluid samples were centrifuged at 1500 rpm for 5 min. The glucose was measured using Cobas C111 analyzer (Roche, USA). The test of mass transfer of glucose out of peritoneum was calculated by using the following formula: (initial dialysate glucose X initial volume)–(final dialysate glucose X final volume) [26].

### Peritoneal membrane histomorphometric analysis

Section of the peritoneal membrane was collected and fixed in Dubosq-Brazil solution for 45 min and then post-fixed in buffered 10% formaldehyde solution. Peritoneal fibrosis was evaluated in sections (3 mm) stained with Masson's Trichrome. At least 10 pictures at 200x magnification were taken from each rat, and the thickness (μm) from all photomicrographs were measured. Then, a mean peritoneal thickness from each rat was calculated. For this procedure, we used digitized images and image analysis software for analysis (Image-Pro Plus Software 7.0, Media Cybernetics, Bethesda, USA).

### Immunohistochemistry

Paraffin-embedded PM sections (4 μm) were incubated with anti-rat α-smooth muscle actin (α-SMA) (Sigma Chemical, St. Louis, USA) and anti-PCNA (DAKO, Glostrup, Denmark). An LSAB-AP System (DAKO) revealed with fast red dye (Sigma), and a NovolinkPolymer Detection System (Leica Microsystems, Newcastle, UK) revealed with diaminobenzidine were employed for antibody detection.

The expressions of α-SMA and PCNA were calculated by the percentage of the positive area relative to the entire field area, using Image-Pro Plus 7.0 software (Media Cybernetics).

### Gene expression analysis

Total RNA was isolated from frozen peritoneal membrane tissues homogenized in 1 mL of Trizol reagent (Invitrogen, Thermo Fisher Scientific, MA, USA), according to the manufacturer's specifications. Quantification of RNA was performed by measuring absorbance at 260 nm in the HITACHI U-2000 Spectrophotometer instrument (Hitachi, USA). Ratios of 260/280 measures ranging from 1.8 to 2.0 were considered satisfactory for purity standards. The first strand of cDNA was synthesized from 0.2 ug of total RNA, using the Moloney Murine Leukemia Virus (M-MLV) transcriptase reverse enzyme (Promega, WI, USA) according to the manufacturer's specifications.

Gene expression of fibronectin, fibroblast-specific protein 1 (FSP-1), vascular endothelial growth factor (VEGF), transforming growth factor beta (TGF-β), SMAD3 and SMAD7 were performed by quantitative real time PCR (qRT-PCR) using SYBR Green detection method. Briefly, the qRT-PCR reactions were conducted in total volume of 20 μL containing 3 μL of diluted cDNA, 1 uL of each primer (10 μM) and 10 μL of SsoFast EvaGreen Supermix 2X (BioRad Laboratories, CA, USA) and water. The primers sequences are sumarized in S1 Table. Primers for β-actin were used as an internal control.

The following PCR cycle conditions were used: 10 minutes at 95˚C, followed by 40 cycles of 15 seconds at 95 ˚C for denaturation, 20 seconds at 60˚C for combined annealing, and 10 seconds at 72˚C for the extension. The reactions were conducted in duplicate on the StepOne Plus instrument (Applied Biosystems, Thermo Fisher).

Data were analysed using the comparative Ct method [27]. Following the method, the mRNA amounts of the target genes were normalized to the endogenous housekeeping β-actin gene. The equation $2^{-\Delta\Delta Ct}$ was applied to calculate the relative gene expression comparing each sample of all studied groups to the mean of a control group where ΔCt = Ct target gene—Ct *β-actin* and ΔΔCt = ΔCt sample—mean ΔCt control group.

## Statistical analysis

Data are presented as mean ± standard error of mean (SEM), and statistical analyses were performed with the GraphPad Prism 5.0 statistical program (GraphPad, San Diego, CA, USA). One-way analysis of variance (ANOVA) with pairwise comparisons according to the Newmann-Keuls formulation was used. A *p*-value less than 0.05 was considered significant.

## Results

### LDE-PXT effect on the function of peritoneal membrane

To determine the effect of LDE-PTX treatment on the preservation of peritoneal function, it was performed the ultrafiltration (UF) rate and the mass transfer of glucose (MTG) analysis. As expected, the peritoneal function was diminished in the PF and LDE groups: a major reduction in the UF rate (Fig 1A) and increase of MTG were observed in the disease model (Fig 1B). On the other hand, treatment with LDE-PTX prevented the decline in the membrane function, as shown in Fig 1: both the UF rate (Fig 1A) and MTG (Fig 1B) were kept at normal levels in the LDE-PTX group.

### LDE-PTX effect on peritoneal membrane fibrosis

The efficiency of LDE-PTX to prevent morphological changes of the peritoneal membrane was tested on a model of peritoneal fibrosis induced by CG injections. As expected, PF (130 ±27 μm) and LDE (95±9.2 μm) groups exhibited a severe membrane thickening compared to the Controls (39±5 μm; p<0.01) (Fig 2). Treatment with LDE-PTX prevented the development of peritoneal fibrosis (40±4.8μm; p<0.01 vs PF and p<0.05 vs LDE) and preserved the monolayer structure, that was similar to that of the Control group (39±5 μm), as shown in Fig 2.

### LDE-PTX effect on myofibroblasts and cellular proliferation in the peritoneal membrane

It is well known that α-SMA is a common marker of myofibroblasts and these cells are present in the fibrosis processes. Accordingly, we examined the expression of α-SMA by immunohistochemistry and it was found a strong expression of α-SMA in both PF and LDE groups compared to the Controls (7.9±1.0% and 7.3±0.5% area, respectively; vs 0±0%; p<0.05) as shown

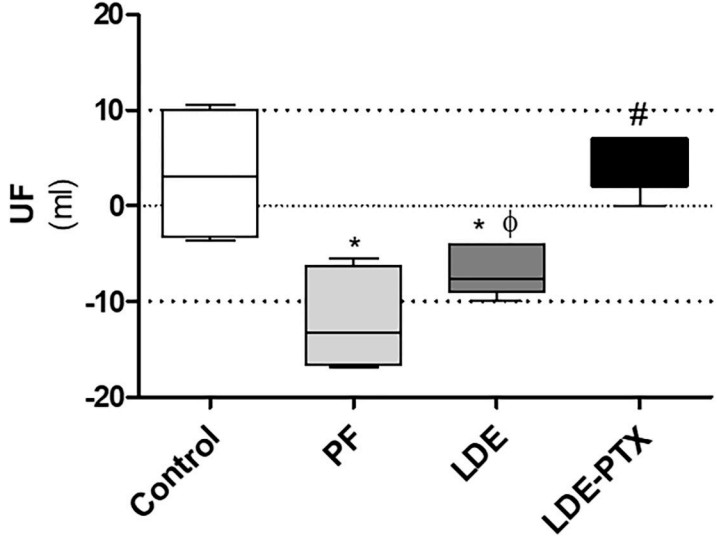

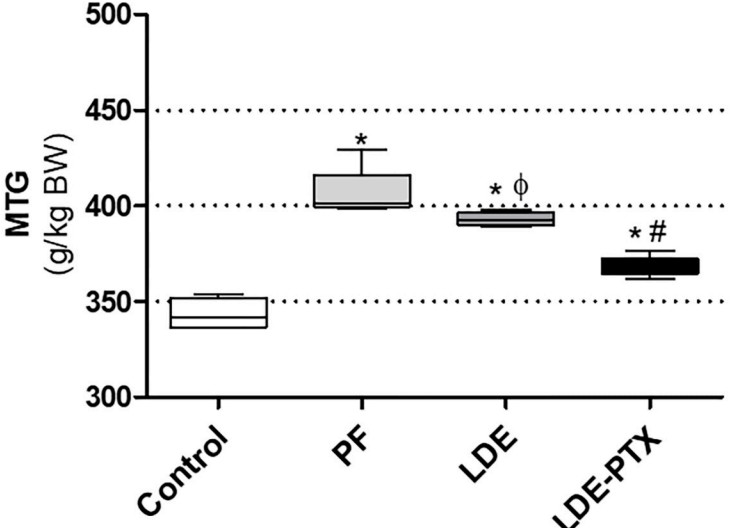

**Fig 1. LDE-PXT treatment on peritoneal function.** The peritoneal fibrosis model (PF) was induced by IP injections of chlorhexidine gluconate (CG) for 15 consecutive days; The LDE group (n = 8), besides receiving CG injections, also received LDE IP injections every 3 days for 15 days; and LDE-PXT group (n = 8) besides receiving CG injections, also received LDE-PTX IP injections (4 mg/kg) every 3 days for 15 days. LDE-PXT treatment preserved the membrane function by conserving the ultrafiltration rate (UF) (**A**) and by maintaining the mass transfer of glucose (MTG) at normal levels (**B**). $^{*}p<0.01$ *vs* Control; $^{\#}p<0.01$ *vs* PF; $^{\Phi}p<0.05$ *vs* LDE-PTX.

in Fig 3. Interestingly, treatment of LDE-PTX reduced the number of myofibroblasts in the peritoneal membrane (3.2±0.1%; p<0.05 vs Control, PF and LDE groups) (Fig 3).

We also evaluated the proliferation of myofibroblasts by PCNA rate to measure the degree of the injury to the peritoneal membrane. As shown in Fig 4 we observed an intense cellular proliferation contributing to the peritoneal membrane thickness in both PF and LDE groups, as compared to the Controls (234±32 and 203±24, respectively vs 1±0.1; p<0.01). In contrast,

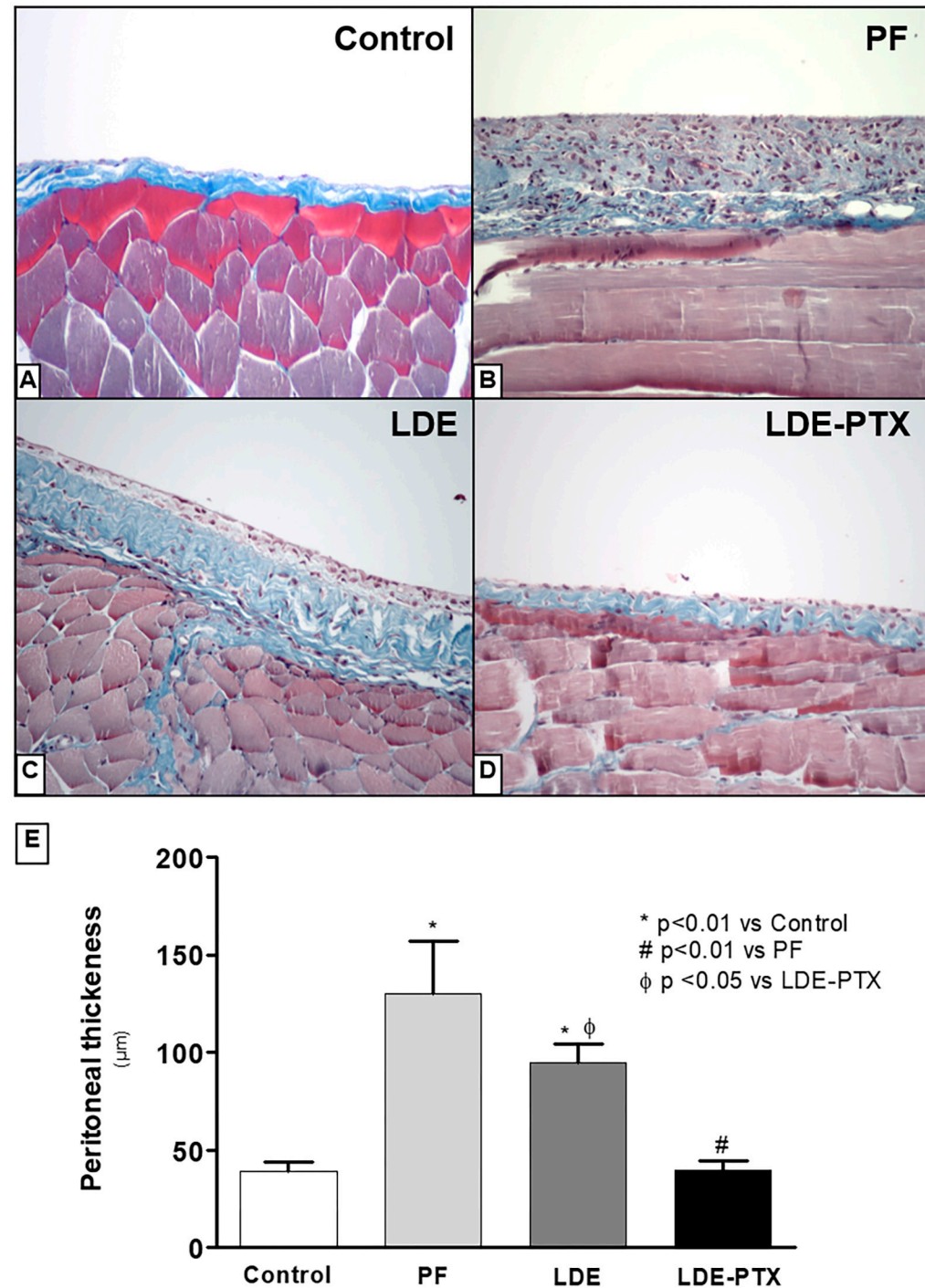

**Fig 2. LDE-PXT treatment on peritoneal thickness.** Representative photomicrographs of peritoneal membrane samples stained with Masson's trichrome (x200). **(A)** The Control group (n = 5) showed a normal peritoneum without any morphological changes. **(B)** The PF group (n = 5) and **(C)** the LDE group (n = 8) showed a significant thickening of the peritoneal membrane. **(D)** LDE-PXT group (n = 8) showed prominent preservation of the peritoneal membrane without fibrosis development and also preserved the monolayer structure at day 15, nearly to the Control group as demonstrated by quantitative analysis **(E)**. $^{*}p<0.01$ *vs* Control; $^{#}p<0.01$ *vs* PF; $^{\Phi}p<0.05$ *vs* LDE-PTX.

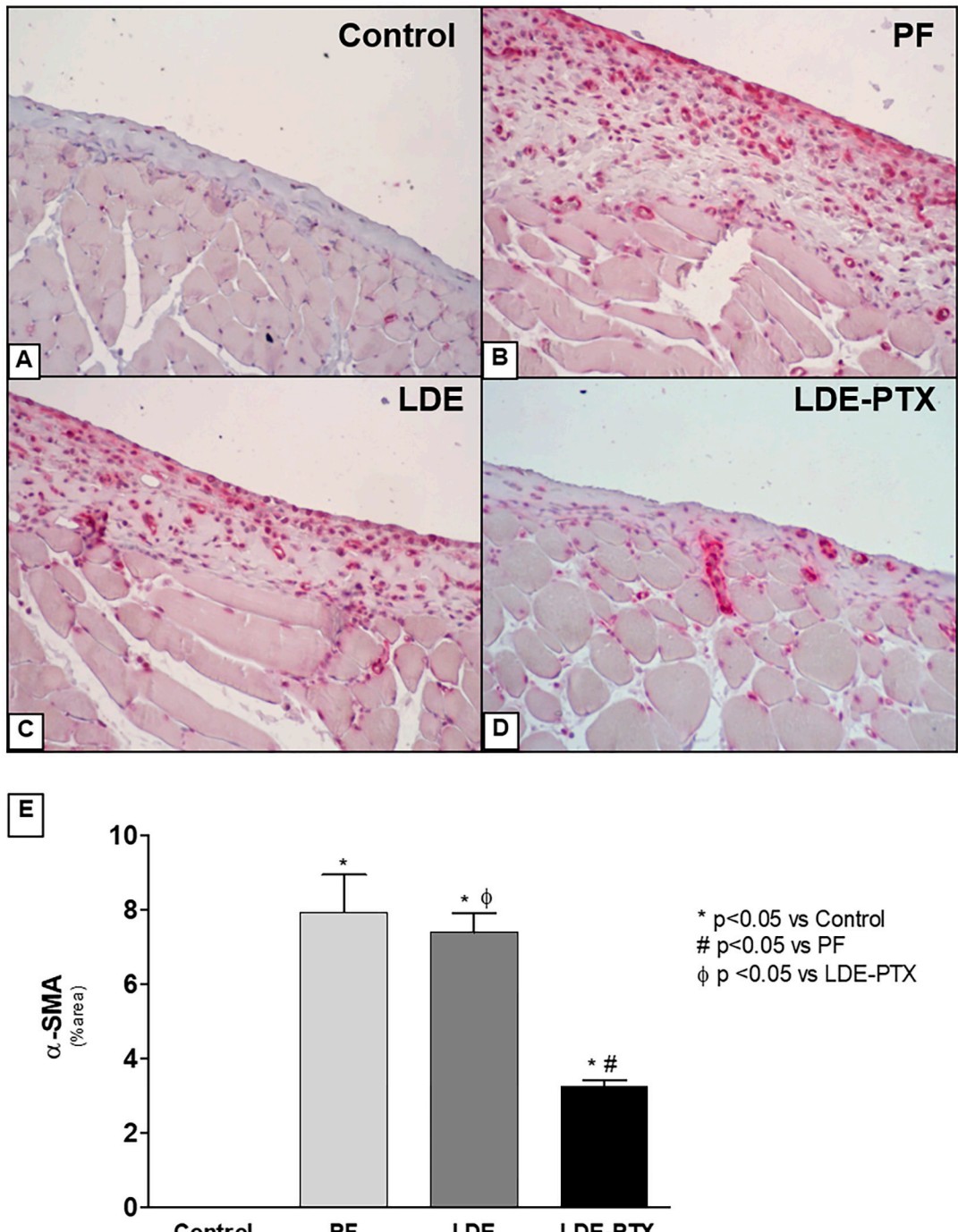

**Fig 3. LDE-PXT treatment on α-SMA expression.** Representative photomicrographs of peritoneal membrane samples stained for anti-rat α-smooth muscle actin (α-SMA) using immunohistochemistry (x200). **(A)** The Control group (n = 5) showed no expression of α-SMA. **(B)** The PF (n = 5) and **(C)** the LDE groups (n = 8) showed a significant increase in α-SMA expression. **(D)** PF+LDE-PXT group (n = 8) exhibited an important reduction of α-SMA expression demonstrated by quantitative analysis **(E)**. $^*p < 0.01$ *vs* Control; $^\#p < 0.05$ *vs* PF; $^\Phi p < 0.05$ *vs* LDE-PTX.

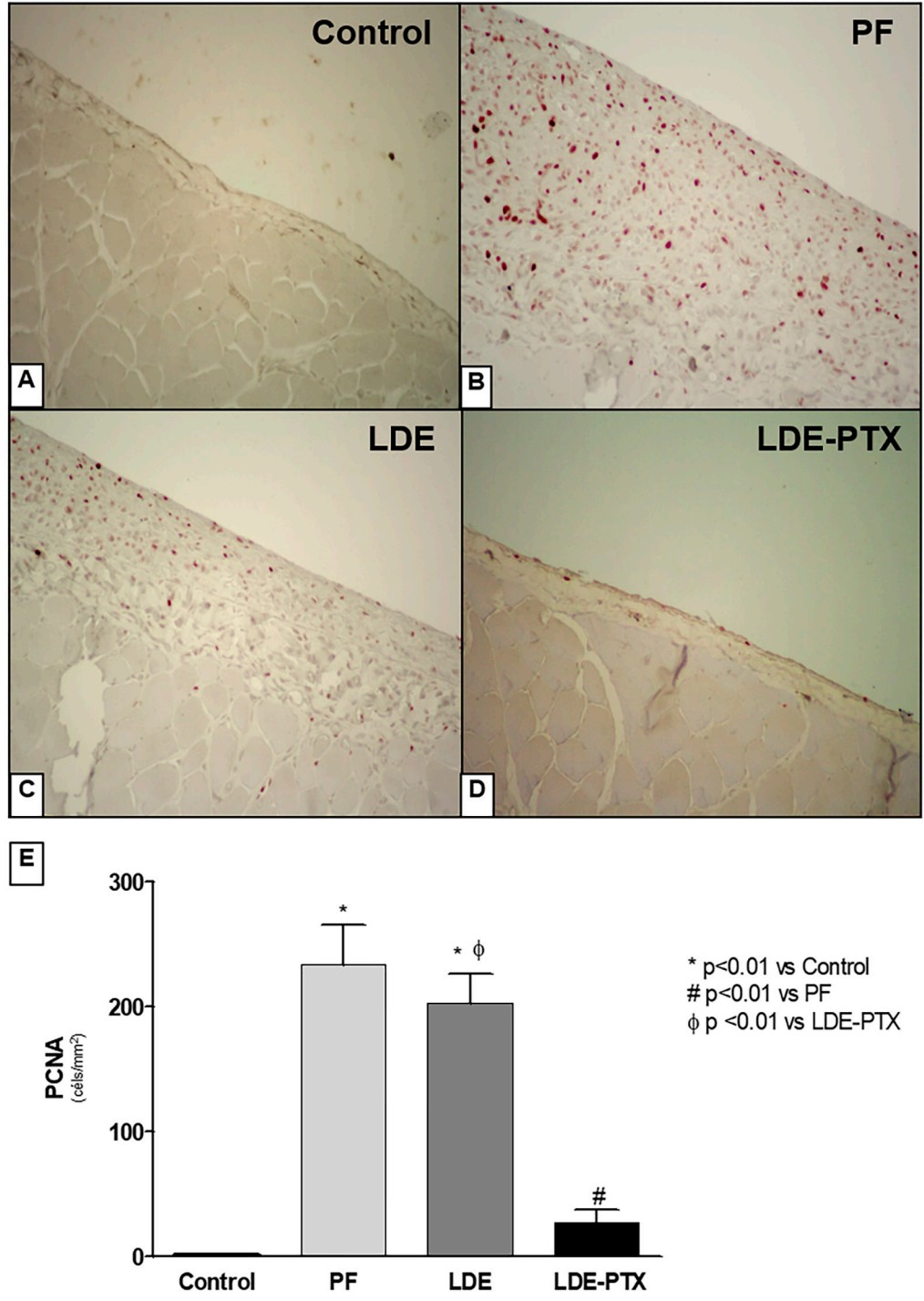

**Fig 4. LDE-PXT treatment on proliferation.** Representative photomicrographs of peritoneal membrane samples stained for anti-proliferation cell nuclear antigen (PCNA) using immunohistochemistry (x200). **(A)** The Control group (n = 5) showed no expression of PCNA. **(B)** The PF (n = 5) and **(C)** the LDE groups (n = 8) showed a significant increase of PCNA expression. **(D)** LDE-PXT group (n = 8) exhibited a significant reduction of PCNA expression demonstrated by quantitative analysis **(E)**. *$p<0.01$ *vs* Control; #$p<0.01$ *vs* PF; $^{\Phi}p<0.01$ *vs* LDE-PTX.

treatment with LDE-PTX effectively blocked the cell proliferation thereby avoiding the peritoneal thickening (27.5± 10; p<0.01 vs PF and LDE groups) (Fig 4).

## LDE-PXT effect on pro-fibrotic cytokines and angiogenesis

To analyze the mechanisms involved in peritoneal fibrosis, we evaluated the gene expression of pro-fibrotic markers such as fibronectin and FSP-1 and also the pro-angiogenesis factor VEGF by qRT-PCR technique. In this model of peritoneal fibrosis, mRNA levels of fibronectin, FSP-1, and VEGF were higher in both PF and LDE groups compared with the Control group (p<0.05, Fig 5). In contrast, animals treated with LDE-PTX reduced the mRNA levels of these markers, as compared to the PF group (p<0.05).

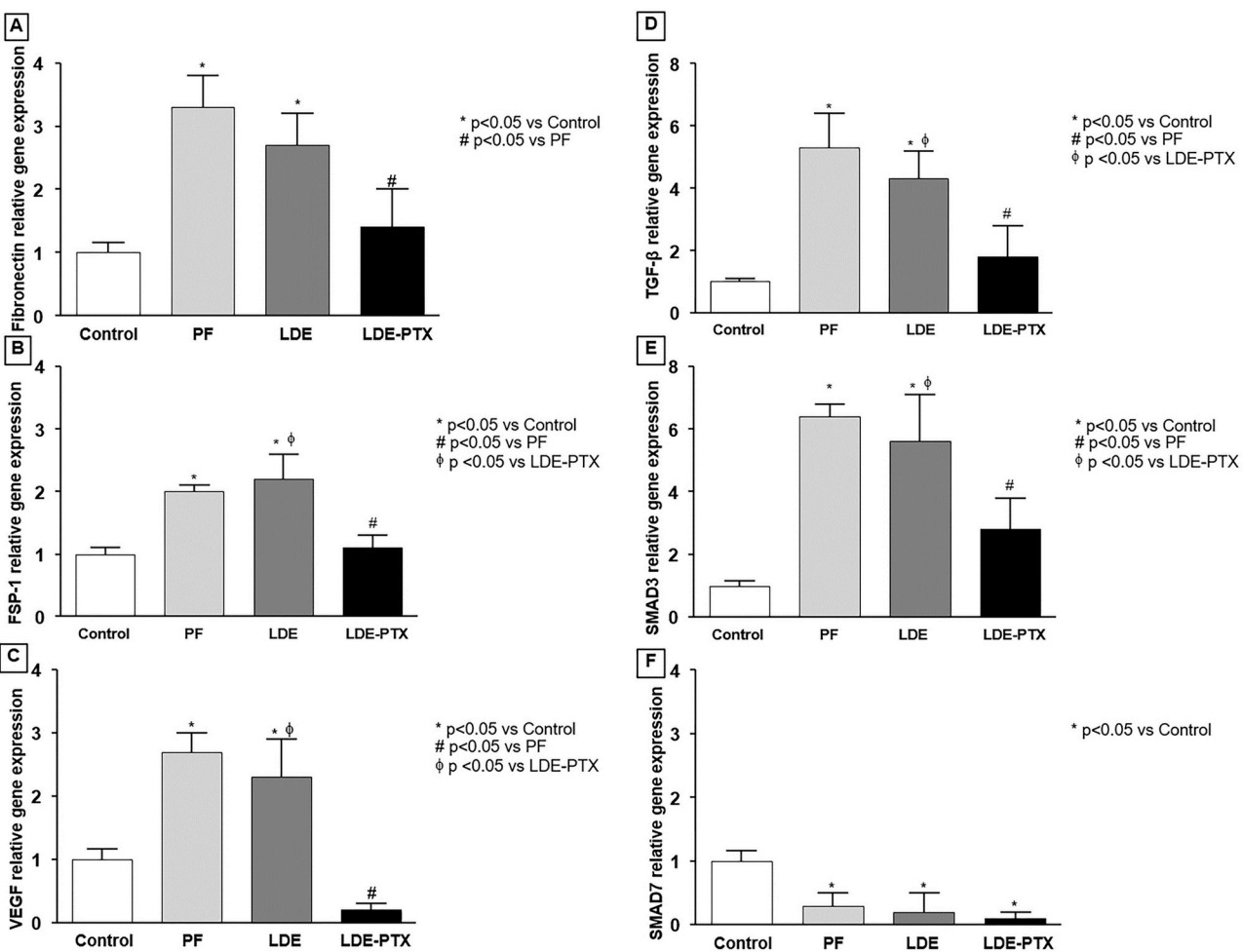

**Fig 5. LDE-PXT treatment on gene expression analysis.** mRNA levels of peritoneal membrane samples were measured by quantitative real-time PCR of **(A)** fibronectin; **(B)** fibroblast-specific protein 1 (FSP-1); **(C)** vascular endothelial growth factor (VEGF); **(D)** transforming growth factor-beta (TGF-β); **(E)** SMAD3; and **(F)** SMAD7. Relative gene expression ($2^{-\Delta\Delta Ct}$) of fibronectin **(A)**, FSP-1 **(B)**, VEGF **(C)**, TGF-β **(D)** and SMAD3 **(E)** were significantly higher in the PF and LDE groups compared to the Control group (*p<0.05). LDE-PXT treatment exhibited an important reduction of mRNA levels of those genes compared to the PF (#p<0.05) and the LDE groups (Φp<0.05). Relative gene expression of SMAD7 **(F)** was down-regulated in the PF, LDE, and LDE-PTX groups compared to the Controls (p<0.05).

## LDE-PXT effect on peritoneal fibrosis via the TGF-β/SMAD pathway

To verify whether treatment with LDE-PTX could also interfere with SMAD expression, which is a well-known intracellular protein involved in the TGF-β signaling. qRT-PCR was performed for the detection of TGF-β, SMAD3, and SMAD7 gene expression. It was found that TGF-β and SMAD3 gene expression were significantly higher in both PF and LDE groups compared to the Control group (p<0.05) as shown in Fig 5. On the other hand, peritoneal TGF-β and SMAD3 gene expression were reduced by LDE-PTX treatment compared to the PF group (p<0.05). In addition, we found down-regulation of SMAD7 expression in the PF, LDE, and LDE-PTX groups compared to the Controls (p<0.05; Fig 5).

## Discussion

In this study, treatment with LDE-PTX of rats with induction of peritoneal fibrosis by CG injection was successful in preventing the development of the disease in this animal model. CG elicits destruction of peritoneal membrane by disrupting the junctions between mesothelial cells, with subsequent damage to the subserous tissue, causing an inflammatory response which in turn favors triggering of pro-fibrotic mechanisms [23]. This model presumably mimics the severe aggression to the peritoneal membrane produced by the continuous sequences of dialysis sessions. Ultimately, fibrosis abolishes the filtering capacity of the peritoneal membrane and the usefulness of the peritoneal dialysis. The clear-cut results obtained here by the potent anti-proliferative agent PTX associated with LDE may shed new light on the quest to overcome fibrosis and warrant the efficacy of peritoneal dialysis for longer periods.

The anti-proliferative action of LDE-PTX on non-neoplastic tissues was previously shown in rabbits with atherosclerosis induced by cholesterol feeding, in which the proliferation of smooth muscle cells in the medium arterial layer with invasion of the intima was prevented [18]. Treatment with another taxane, docetaxel, as associated with LDE also prevented smooth muscle cells proliferation and intimal invasion, as expected from the LDE-PTX results from the atherosclerotic rabbits [28]. Treatment with LDE-docetaxel reduced the protein expression of the proliferation marker PCNA by 40% compared to controls [28]. Here, LDE-PTX reduced myofibroblast proliferation at similar rates, as expressed by the α-SMA and PCNA markers. Thus, our current findings obtained from peritoneal myofibroblast recruitment confirm the wide action of taxanes as blockers of cell proliferation in other processes beyond cancer and atherosclerosis.

In our study, as expected, the induction of fibrosis in rats promoted the increase of gene expression of the pro-fibrotic markers fibronectin, FSP-1, TGF-β, SMAD3 and VEGF, that favours angiogenesis and thereby assists the perfusion of the newly formed fibrotic tissue [12, 13, 19]. It is noteworthy that LDE-PTX treatment pronouncedly diminished the gene expression of all those molecular factors, thus interfering with diverse mechanisms of fibrogenesis. This can be accounted for the success of the LDE-PTX in achieving the prevention of fibrosis appearance in the treated rats.

SMAD7 is an inhibitor of the TGF-β signal transduction that promotes fibrogenesis [29]. Given our results, it would be then expected that SMAD7 would be overexpressed. In fact, the induction of fibrosis by CG resulted in decreasing in SMAD7 expression and LDE-PTX treatment did not trigger a stimulatory response that would inhibit TGF-β. Lack of response of SMAD7 to LDE-PTX suggests that the TGF-β/SMAD3 pathway was inhibited by alternative mechanisms.

Treatment with LDE-docetaxel lowered the protein expression of other participants of the fibrosis process, such as collagen, metalloproteinases and also TGF-β in rabbits [28]. Those

findings highlight the wide capacity of taxanes carried in LDE to lessen fibrosis development in different animal species and different disease models.

In this rat model of peritoneal fibrosis induction used in this study, LDE-PTX treatment commenced simultaneously with the daily injections of CG. Thus, our current results point out the success of LDE-PTX in preventing damage to the filtration capacity of the peritoneal membrane. Perhaps a more defying experimental challenge would consist at the beginning of the treatment when peritoneal fibrosis would be fully developed, but this approach might bring considerable experimental difficulty. Some objections have been posed to the CG model for PF induction in rats as to mirror PF in the human clinical setting, particularly for lacking of the renal failure component [30]. At any rate, the beneficial effects of LDE-PTX clearly involved the fibrotic and inflammatory bases of the disease, with applicability in either preventive or therapeutic scenarios.

The conventional formulation of PTX, in which the compound is dissolved in Chremophor L bears important toxicities, with prominent myelotoxicity, specially neutropenia, and other toxicities such as peripheral neuropathy and myalgias. Renal toxicity is not related to PTX, which is excreted via bile. It is remarkable that, when associated with LDE, PTX toxicity is practically diminished: the 50% lethal dose (LD50) of LDE-PTX is roughly tenfold higher than that of conventional PTX [16]. Observable toxicity was nearly absent in experimental animal models of atherosclerosis [18, 31], cancer [16, 32], heart [33], and aorta [19] transplantation, in mice, rats, rabbits and monkeys [34]. In clinical trials enrolling patients with advanced cancers [17, 20, 21] and cardiovascular disease [22], no clinical or laboratorial toxicities were observed at the 175 mg/m$^2$ body surface triweekly dose used in those studies. Thus, it is reasonable to expect that patients with renal failure would not present toxicities related with LDE-PTX treatment. Thus, the safety and near absence of clinical and laboratory toxicity of LDE-PTX for human use, as shown in our previous studies, opens the avenue for future clinical studies designed in settings of either prevention or treatment of established peritoneal fibrosis.

In conclusion, our results suggest that LDE-PTX has the potential to be tested as adjuvant therapy to peritoneal dialysis, aiming prevention of peritoneal fibrosis caused by this procedure. The fact that in previous clinical studies LDE-PTX has been shown to be devoid of observable toxicity further supports the feasibility of this novel approach.

## Supporting information

**S1 Table. Primer sequences used for qRT-PCR.**
(DOCX)

## Author Contributions

**Conceptualization:** Filipe M. O. Silva, Raul C. Maranhão, Irene L. Noronha.

**Data curation:** Filipe M. O. Silva, Priscila O. Carvalho, Elerson C. Costalonga, Rafael Pepineli, Raul C. Maranhão, Irene L. Noronha.

**Formal analysis:** Filipe M. O. Silva, Priscila O. Carvalho, Elerson C. Costalonga, Rafael Pepineli, Irene L. Noronha.

**Funding acquisition:** Raul C. Maranhão.

**Investigation:** Filipe M. O. Silva, Priscila O. Carvalho, Elerson C. Costalonga, Rafael Pepineli, Irene L. Noronha.

**Methodology:** Filipe M. O. Silva, Priscila O. Carvalho, Elerson C. Costalonga, Rafael Pepineli.

**Project administration:** Filipe M. O. Silva.

**Resources:** Filipe M. O. Silva, Priscila O. Carvalho, Raul C. Maranhão.

**Software:** Filipe M. O. Silva, Priscila O. Carvalho, Elerson C. Costalonga.

**Supervision:** Raul C. Maranhão, Irene L. Noronha.

**Validation:** Filipe M. O. Silva, Priscila O. Carvalho, Irene L. Noronha.

**Visualization:** Filipe M. O. Silva, Priscila O. Carvalho, Raul C. Maranhão, Irene L. Noronha.

**Writing – original draft:** Filipe M. O. Silva, Priscila O. Carvalho, Raul C. Maranhão, Irene L. Noronha.

**Writing – review & editing:** Filipe M. O. Silva, Priscila O. Carvalho, Elerson C. Costalonga, Raul C. Maranhão, Irene L. Noronha.

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
