## [Decision Letter · Decision Letter 0]

7 Jan 2022

PONE-D-21-37828USE OF PACLITAXEL CARRIED IN SOLID LIPID NANOPARTICLES TO PREVENT PERITONEAL FIBROSIS IN RATSPLOS ONE

Dear Dr. Maranhão,

Thank you for submitting your manuscript to PLOS ONE. After careful consideration, we feel that it has merit but does not fully meet PLOS ONE’s publication criteria as it currently stands. Therefore, we invite you to submit a revised version of the manuscript that addresses the points raised during the review process.

ACADEMIC EDITOR:

Two expert reviewers have evaluated this MS which is well written and relevant, and the study appears well designed and executed. Some concerns will need to be addressed regarding the validity of the model chosen and how the data would translate into the clinical arena, and a few specific points below. Happy to receive the revised MS for re-evaluation regarding suitability for publication in PlosOne.

We look forward to receiving your revised manuscript.

Kind regards,

Frank JMF Dor, M.D., Ph.D., FEBS, FRCS

Academic Editor

PLOS ONE

Journal Requirements:

2. To comply with PLOS ONE submissions requirements, in your Methods section, please provide additional information on the animal research and ensure you have included details on (1) basic housing and breeding, (2) health monitoring, and (3) efforts to alleviate suffering.

3. As part of your revision, please complete and submit a copy of the Full ARRIVE 2.0 Guidelines checklist, a document that aims to improve experimental reporting and reproducibility of animal studies for purposes of post-publication data analysis and reproducibility: https://arriveguidelines.org/sites/arrive/files/Author%20Checklist%20-%20Full.pdf (PDF). Please include your completed checklist as a Supporting Information file. Note that if your paper is accepted for publication, this checklist will be published as part of your article.

"This study was supported by the State of Sao Paulo Research Support Foundation (FAPESP, Sao Paulo, Brazil, Grant number 2014/03742-0) and by the National Institute of Science and Technology for Complex Fluids of the Ministry of Science, Technology and Innovation of Brazil."

Reviewers' comments:

Reviewer's Responses to Questions

**Comments to the Author**

1. Is the manuscript technically sound, and do the data support the conclusions?

Reviewer #1: Yes

Reviewer #2: Yes

2. Has the statistical analysis been performed appropriately and rigorously? 

Reviewer #1: Yes

Reviewer #2: I Don't Know

3. Have the authors made all data underlying the findings in their manuscript fully available?

Reviewer #1: No

Reviewer #2: Yes

4. Is the manuscript presented in an intelligible fashion and written in standard English?

Reviewer #1: Yes

Reviewer #2: Yes

5. Review Comments to the Author

Reviewer #1: Thank you for the opportunity to review this work his is a well designed and conducted study investigating the effect of paclitaxel within lipid nanoparticles to mitigate cholhexidine-gluconate induced peritoneal fibrosis.

While this work has been well designed and conducted the key question relates to whether this has clinical relevance. Specifically, there is limited evidence that chlorhexidine-gluconate is a relevant model of PD-induced peritoneal fibrosis see Vlijm et al 2011 Nephron Exp Nephrol.

Could the authors comment on why they think that LDE without paclitaxel appears to have 50% as compared with LDE+paclitaxel in mitiigating fibrosis?

Please review this sentence

"Conversely, the less expensive, home-based PD procedure, which is more often used in poor countries, has important shortcomings". LMIC (lower and middle income countries) may be a more specific

description

Reviewer #2: This study investigates the effect of intraperitoneal Paclitaxel on the formation of peritoneal fibrosis following exposure to chlorhexidine gluconate. The topic is important as a sizeable number of patients undergoing Peritoneal dialysis must go over to hemodialysis due to the loss of ultrafiltration ability of peritoneum secondary to fibrosis. The study design is adequate and the experiment appears well performed as the authors covered many relevant aspects. This reviewer would have liked to hear briefly about the potential side effects of Paclitaxel or if the results would have been different if the rats had some sort of kidney impairment.

6. PLOS authors have the option to publish the peer review history of their article (what does this mean?). If published, this will include your full peer review and any attached files.

Reviewer #1: No

Reviewer #2: No

---

## [Author Response · Author response to Decision Letter 0]

24 Feb 2022

Journal Requirements:

- Ok, the requirements were met.

2. To comply with PLOS ONE submissions requirements, in your Methods section, please provide additional information on the animal research and ensure you have included details on (1) basic housing and breeding, (2) health monitoring, and (3) efforts to alleviate suffering.

- The additional information on the animal research was included in the Methods section (page 6, lines 121-126).

3. As part of your revision, please complete and submit a copy of the Full ARRIVE 2.0 Guidelines checklist, a document that aims to improve experimental reporting and reproducibility of animal studies for purposes of post-publication data analysis and reproducibility: https://arriveguidelines.org/sites/arrive/files/Author%20Checklist%20-%20Full.pdf (PDF). Please include your completed checklist as a Supporting Information file. Note that if your paper is accepted for publication, this checklist will be published as part of your article.

- Ok, the requirements were met.

"This study was supported by the State of Sao Paulo Research Support Foundation (FAPESP, Sao Paulo, Brazil, Grant number 2014/03742-0) and by the National Institute of Science and Technology for Complex Fluids of the Ministry of Science, Technology and Innovation of Brazil."

Reply to the reviewers:

Beforehand, we would like to deeply thank both Reviewers for their excellent work that helped us to improve our manuscript. 

Reviewer #1: 

Thank you for the opportunity to review this work his is a well designed and conducted study investigating the effect of paclitaxel within lipid nanoparticles to mitigate cholhexidine-gluconate induced peritoneal fibrosis.

1. While this work has been well designed and conducted the key question relates to whether this has clinical relevance. Specifically, there is limited evidence that chlorhexidine-gluconate is a relevant model of PD-induced peritoneal fibrosis see Vlijm et al 2011 Nephron Exp Nephrol.

- We are now mentioning other PD animal models and justifying our choice of the chlorhexidine-gluconate model for this study (See Introduction section, with two references included, number 23 and 24, page 5, lines 102-111), with further comments in the Discussion section (ref. number 23, page 16, lines 356-359 of the revised manuscript). In respect to the Vlijm et al (2011) paper, which is now being quoted (ref. number 30, page 18, line 405-407 of the revised manuscript).

2. Could the authors comment on why they think that LDE without paclitaxel appears to have 50% as compared with LDE+paclitaxel in mitiigating fibrosis?

- As showed in Figure 2, the difference observed in the peritoneal thickness of PF group compared with LDE group was not statistically significant. Thus, we assumed that LDE did not have any effect on peritoneal fibrosis, as expected.

3. Please review this sentence

"Conversely, the less expensive, home-based PD procedure, which is more often used in poor countries, has important shortcomings". LMIC (lower and middle income countries) may be a more specific description

- Ok, done (Introduction section, page 3, line 67).

Reviewer #2: 

This study investigates the effect of intraperitoneal Paclitaxel on the formation of peritoneal fibrosis following exposure to chlorhexidine gluconate. The topic is important as a sizeable number of patients undergoing Peritoneal dialysis must go over to hemodialysis due to the loss of ultrafiltration ability of peritoneum secondary to fibrosis. The study design is adequate and the experiment appears well performed as the authors covered many relevant aspects. 

1. This reviewer would have liked to hear briefly about the potential side effects of Paclitaxel or if the results would have been different if the rats had some sort of kidney impairment.

- See comments respecting those issues in the Discussion section (page 19, lines 411-424). We also included 4 papers of our previous studies regarding this issue (see reference list, number 31-34, pages 26-27 of the revised manuscript).

---

## [Decision Letter · Decision Letter 1]

25 Apr 2022

Use of paclitaxel carried in solid lipid nanoparticles to prevent peritoneal fibrosis in rats

PONE-D-21-37828R1

Dear Dr. Maranhão,

We’re pleased to inform you that your manuscript has been judged scientifically suitable for publication and will be formally accepted for publication once it meets all outstanding technical requirements.

Kind regards,

Frank JMF Dor, M.D., Ph.D., FEBS, FRCS

Academic Editor

PLOS ONE

Additional Editor Comments (optional):

Reviewers' comments:

Reviewer's Responses to Questions

**Comments to the Author**

1. If the authors have adequately addressed your comments raised in a previous round of review and you feel that this manuscript is now acceptable for publication, you may indicate that here to bypass the “Comments to the Author” section, enter your conflict of interest statement in the “Confidential to Editor” section, and submit your "Accept" recommendation.

Reviewer #1: All comments have been addressed

Reviewer #2: All comments have been addressed

2. Is the manuscript technically sound, and do the data support the conclusions?

Reviewer #1: Yes

Reviewer #2: (No Response)

3. Has the statistical analysis been performed appropriately and rigorously? 

Reviewer #1: I Don't Know

Reviewer #2: (No Response)

4. Have the authors made all data underlying the findings in their manuscript fully available?

Reviewer #1: No

Reviewer #2: (No Response)

5. Is the manuscript presented in an intelligible fashion and written in standard English?

Reviewer #1: Yes

Reviewer #2: Yes

6. Review Comments to the Author

Reviewer #1: Thank you all questions have been addressed to my satisfaction.

I think the clinical relevance of this work remains unclear.

Reviewer #2: (No Response)

7. PLOS authors have the option to publish the peer review history of their article (what does this mean?). If published, this will include your full peer review and any attached files.

Reviewer #1: No

Reviewer #2: No

---

## [Editor Report · Acceptance letter]

29 Apr 2022

PONE-D-21-37828R1 

Use of paclitaxel carried in solid lipid nanoparticles to prevent peritoneal fibrosis in rats 

Dear Dr. Maranhão:

I'm pleased to inform you that your manuscript has been deemed suitable for publication in PLOS ONE. Congratulations! Your manuscript is now with our production department. 

Kind regards, 

on behalf of

Dr. Frank JMF Dor 

Academic Editor

PLOS ONE